# Processing of Highly Filled Polymer–Metal Feedstocks for Fused Filament Fabrication and the Production of Metallic Implants

**DOI:** 10.3390/ma13194413

**Published:** 2020-10-03

**Authors:** Christopher Gloeckle, Thomas Konkol, Olaf Jacobs, Wolfgang Limberg, Thomas Ebel, Ulrich A. Handge

**Affiliations:** 1Department of Mechanical Engineering and Business Administration, Technische Hochschule Lübeck, Mönkhofer Weg 239, 23562 Lübeck, Germany; christopher.gloeckle@hzg.de (C.G.); tomkonkol@comcast.net (T.K.); olaf.jacobs@th-luebeck.de (O.J.); 2Institute of Materials Research, Helmholtz-Zentrum Geesthacht, Max-Planck-Strasse 1, 21502 Geesthacht, Germany; wolfgang.limberg@hzg.de (W.L.); thomas.ebel@hzg.de (T.E.); 3Institute of Polymer Research, Helmholtz-Zentrum Geesthacht, Max-Planck-Strasse 1, 21502 Geesthacht, Germany

**Keywords:** fused filament fabrication, polymer–metal composites, rheology, sintering, metallic implants

## Abstract

Fused filament fabrication (FFF) is a new procedure for the production of plastic parts, particularly if the parts have a complex geometry and are only needed in a limited quantity, e.g., in specific medical applications. In addition to the production of parts which are purely composed of polymers, fused filament fabrication can be successfully applied for the preparation of green bodies for sintering of metallic implant materials in medical applications. In this case, highly filled polymer–metal feedstocks, which contain a variety of polymeric components, are used. In this study, we focus on various polymer-metal feedstocks, investigate the rheological properties of these materials, and relate them to our results of FFF experiments. Small amplitudes of shear oscillations reveal that the linear range of the polymer–metal feedstocks under investigation is very small, which is caused by elastic and viscous interactions between the metallic particles. These interactions strongly influence or even dominate the flow properties of the feedstock depending on the applied shear stress. The magnitude of the complex viscosity strongly increases with decreasing angular frequency, which indicates the existence of an apparent yield stress. The viscosity increase caused by the high powder loading needed for sintering limits the maximum printing velocity and the minimum layer height. The apparent yield stress hinders the formation of smooth surfaces in the FFF process and slows down the welding of deposited layers. The influence of composition on the processing parameters (suitable temperature range) and part properties (e.g., surface roughness) is discussed on the basis of rheological data.

## 1. Introduction

Metallic implants are frequently used in medical applications. Typical examples are medical stents, bone plates, and screws, as well as joint implants. In addition to conventional processing like forging and machining, sintering is applied for the fabrication of these implants in an increasing number of applications [1,2,3]. The motivation is either achievement of special material properties or application of modern shaping technologies related to the usage of fine metallic powders. These technologies provide the possibility to individualize implants to the needs of distinct patients, e.g., bone plates for fracture stabilization. These processes usually require a so-called feedstock material with a sufficiently high volume fraction of metallic powder particles which is mixed with a binder consisting of different polymers. A green body can be prepared from this feedstock, e.g., using the method of metal injection molding (MIM) [4,5]. Injection molding is typically associated with a high pressure on the order of several hundred bar during injection. Furthermore, a special mold, which is usually quite expensive, is needed for the MIM process. After injection molding, the polymers are typically removed from the green part by chemical or solvent debinding and a subsequent thermal process, after which the powder is sintered to the final pure metallic component. High pressures and the use of a mold both lead to some advantageous properties, including a smooth surface of the green body and accurate geometrical dimensions. Voids can also be avoided to a large extent in the MIM process. Metal injection molding has been an established cost-effective production process for several decades, and it is typically used to manufacture large volumes of small complex-shaped parts. However, for small quantities, the high expenses for the mold make the technology less attractive. Therefore, other techniques adapted from additive manufacturing technologies related to the usage of feedstock are under development to enable the shaping of single parts or small series [6,7,8,9,10].

Recently, fused filament fabrication (FFF) was used to prepare green bodies for sintering [11,12,13,14,15]. In contrast to MIM, where the feedstock is provided as granules, here, the feedstock material is needed in a filament form. In comparison to the equipment used for the MIM process, the costs for installation of the FFF equipment are much lower. In addition, a nearly unlimited variety of geometries can be achieved using the FFF process without the design of an expensive mold. However, processing of feedstock materials is much more challenging than printing pristine polymers which are typically used for fused filament fabrication, since the feedstock materials need to have a sufficiently high fraction of metallic powder particles. Consequently, only a very small number of polymers filled with metal powder particles are currently commercially available, with commercial products mainly focusing on stainless-steel powders. In addition to optimum flow properties, feedstock filaments for the FFF process need to possess a certain degree of flexibility such that they can be spooled and bent easily. Furthermore, possible detrimental reactions between binder residuals and the metallic powders during thermal debinding and sintering have to be considered [16]. This is especially true for highly reactive materials like titanium and magnesium [17]. Therefore, a variety of conditions have to be fulfilled such that a feedstock can be successfully used for both fused filament fabrication and subsequent sintering of a metallic part.

The use of filaments made from filled polymers for metallic implants is a relatively novel approach for the preparation of green bodies. Nevertheless, the quality of printed parts (e.g., their smoothness) has to be increased and the reproducibility of the process has to be improved (see Figure 1). The FFF process can be divided into three main steps associated with different thermal and deformation rate conditions: (i) the feeding of the solid filament into the liquefier, (ii) the flow of the feedstock melt in the nozzle, and (iii) the deposition and subsequent solidification of the feedstock melt on the existing layer of the printed part. An overview of the process design and modeling was given by Turner et al. [18]. In the first step (i.e., the feeding of the filament into the liquefier), buckling of the filament may occur. A buckling criterion was derived by Venkataram et al. [19] which considers the ratios of tensile modulus and viscosity of the used polymer. The melt flow in the nozzle was thoroughly described in the work of McIlroy and Olmsted [20]. On the contrary, only a few studies focused on the last step, i.e., the deposition and solidification of the extruded melt.

Generally, binder and feedstock materials consist of several components which are necessary for fused filament fabrication and sintering, serving as shaping aids and mechanical stabilizers through the different steps of the processing chain. It is well known that the rheological properties of polymeric materials influence their behavior in processing, e.g., phenomena like strain hardening in melt elongation and shear thinning [21]. Consequently, several investigations are devoted to the influence of the rheological properties of pristine polymers on their processing behavior in fused filament fabrication. The failure feed velocity that limits the maximum extrusion rate was determined by Mackay et al. [22]. In the work of Phan and coauthors [23], a model taking into account rheological and heat transfer effects in the nozzle flow of an FFF printer was analyzed. Their work quantified the heat transfer effects limiting the manufacturing rate. A recent overview of this topic with an emphasis on pristine polymer melts was given in the review of Mackay [24]. In addition, modeling studies were performed, showing that the deposition process disentangles the polymer melt [20]. Furthermore, in-line rheological methods were applied in order to monitor the FFF process, and their validity for characterization of new materials for FFF printing was shown [25]. A variety of studies focused on the improvement of different polymeric materials for three-dimensional (3D) printing, e.g., the work by Nguyen et al. [26], which used carbon fibers to improve the mechanical properties of the printed part. In a recent publication, a novel approach was presented to evaluate feedstock compositions for metal fused filament fabrication [27]. The authors stated that the shear force that was estimated on the basis of the shear strength of the filament was the key parameter determining successful printing. In the work of Seppala and Migler [28], the decrease in weld temperature was experimentally determined. Their work showed that the melt stays above the glass transition temperature for only approximately 1 s.

Despite these studies, there are still many open questions in practice. In addition to purely technical issues such as reliable transport of the filament by the printer, heat transfer, sticking on the build plate, and further issues, the complexity of binder composition, powder size distribution, powder surface, and chemical reactions with the metal make the prediction of the optimal filament properties and binder composition very difficult. In practice, changing one component usually means a need for a comprehensive set of experimental trials. Therefore, more basic research is needed in all steps of the metal FFF process. This work provides a first step in this direction by aiming to link the rheological and printing properties of feedstocks that are of relevance for medical implants. Therefore, thermal analysis, rheological experiments, and analysis of printing behavior were performed in this work.

In this work, we focus on the flow and printing properties of selected feedstock materials, i.e., their properties in the melt state. Several in-house designed feedstock materials with different amounts of titanium particles were used for the experimental investigations, and a practical ring geometry for analysis of FFF printing properties was taken. Titanium was selected because of its relevance for medical implants. In addition, a standard 3D printing material, a grade of the linear homopolymer polylactide, was chosen as a reference material. All feedstock materials fulfilled the criteria of printability using FFF, and they were deemed debindable and sinterable if the volume concentration of titanium particles was equal or larger than 50%. The objective of this study was to evaluate the flow properties of feedstock materials for optimum processing.

## 2. Materials and Methods

In this study, a commercial homopolymer and a specially designed series of feedstocks were chosen for the experimental investigations. A pristine polylactide (PLA, REC, Moscow, Russia) was used as a standard commercial reference material for fused filament fabrication. Furthermore, a series of feedstocks with varying amounts of titanium particles were prepared for a systematic analysis. The binder components are listed in Table 1 and consist of a poly(propylene–ethylene) copolymer composed of isotactic propylene repeat units and a random ethylene distribution, a poly(ethylene–vinyl acetate) copolymer, poly(isobutene), and stearic acid. The volume concentration of the titanium particles ranged from 0% to 60% in increments of 10%. In addition, a feedstock with 65 vol.% titanium particles, which is the standard amount of MIM feedstocks and also close to the critical loading for spherical particles, was also prepared.

The chosen metal essentially determines the properties of the final part and, thus, plays a dominant role in processing. For the in-house prepared series of feedstocks, gas-atomized spherical Ti-6Al-4V alloy powder (density 4.5 g/cm^3^, particle diameter <20 μm, Advanced Powders & Coatings, Boisbriand, QC, Canada) was chosen, hereinafter referred to as titanium in general for ease of reading. Generally, titanium is characterized by a high strength and biocompatibility.

In order to achieve a complete evaluation of the mixing process, the feedstock properties, and printing behavior, a comprehensive material characterization was undertaken. Before the tests, all samples were stored under vacuum at a temperature of 30 °C. Thermalgravimetric analysis (TGA) was performed using a TG 209 F1 Libra^®^ (Netzsch, Selb, Germany). The heating rate was 10 K/min, and an argon atmosphere was applied. Differential scanning calorimetric (DSC) experiments were carried out using a DSC 1 Star^e^ System (Mettler-Toledo, Greifenberg, Switzerland). This heating rate was also 10 K/min. The experiments were performed in a nitrogen atmosphere. A heating–cooling–heating cycle was applied. The first heating cycle ranged from room temperature to a temperature 60 °C above the highest melting temperature *T_m_* of the material. The subsequent cooling run went to –150 °C, and the final heating cycle ranged from −150 °C to *T_m_* 60 °C. The thermal transitions were determined on the basis of the second heating cycle.

Samples for the rheological tests were prepared by means of compression molding. The cylindrical samples had dimensions of 8 mm diameter by 2 mm thickness (PLA) or 14 mm diameter by 1 mm thickness for the in-house designed feedstocks. Before compression molding, the samples were dried under vacuum at a temperature of 30 °C. The samples were created via a hot-press compression molding device (Paul-Otto Weber, Remshalden, Germany). The hot press applies a high compressive force of approximately 60 kN. The compression molding temperature varied for the different materials (200 °C for PLA and 120 °C for the in-house designed feedstocks).

Microscopic investigations of the titanium powder were performed on a “Tescan Vega3 SB” (TESCAN, Brno, Czech Republic) scanning electron microscope (SEM) in backscattered (BSE) mode using a work distance of 10 mm and an accelerating voltage of 13 kV. Scanning electron microscopy investigations of the feedstock materials were completed to get a rough impression for the homogeneity of compression-molded samples with respect to the powder particle distribution within the polymer binder matrix. This imaging was performed using a scanning electron microscope (Merlin, Zeiss, Oberkochen, Germany) in BSE mode. The acceleration voltage was 3.0 kV. For investigations of the cross-sections, the samples were fractured in liquid nitrogen.

A blend of polymers and additives, called a binder material, was used and mixed with the metal powder in order to facilitate processing of the pristine metal powder. To evaluate the dependence of rheological and printing properties on the powder load, eight different feedstocks were prepared using titanium powder, i.e., with 0, 10, 20, 30, 40, 50, 60, and 65 vol.% metal powder.

The process of mixing the metal powder with the binder material utilized a planetary mixer (THINKY Inc., Laguna Hills, CA, USA) and either a hot plate or an oven. First, the primary polymers and nonprimary polymers were mixed together in a metal mixing canister (Table 1). The mass of each component was measured as it was mixed to ensure the proper ratios. Then, air was removed from the canister before it was placed inside a glove-box system (M. BRAUN INERTGAS-SYSTEME GMBH, Garching, Germany) with controlled argon atmosphere, in which the mass of the metal powder could be measured and mixed. The canister was subsequently covered and brought to the oven (Memmert, Schwabach, Germany), where it was heated above the melting temperature of the polymers. Afterward, the canister was removed from the oven and placed into a planetary mixer where powder and binder were mixed for 5 min until the blend appeared smooth and homogeneous. The homogenous melt was cooled and subsequently ground into pellets to be used in the extruder. The extruder (Collin, Teach Line E 20 T, COLLIN Lab & Pilot Solutions GmbH, Maitenbeth, Germany) was used in tandem with a conveyer belt to create the desired filament with a diameter of approximately 2.7 mm. The *L*/*D* ratio of the extruder was 25, and the diameter of the cylindrical die was 3 mm.

Rings with a target geometry consisting of a 40 mm outer diameter, a wall thickness of 0.8 mm, and a height of 4 mm were produced on the Ultimaker 3 (Ultimaker B.V., Utrecht, the Netherlands) printer using a nozzle diameter of 0.8 mm (see Figure 2). The ring geometry was chosen as a well-defined test geometry which can be printed with constant flow rate, mimics thin elements of implants because of its thin wall structure, and provides a few benefits. A simple geometry is easy to measure (diameter, height, and wall thickness) and large enough that any changes in volume can be easily detected. The thin wall also makes it easy to observe surface issues. In addition, the print can be produced via a continuous extrusion process. If an untested filament was loaded into the printer, then initial printing tests were completed by holding all printing parameters constant and only changing one variable at a time. The most important parameters were determined to be nozzle temperature, layer height, flow, and printing velocity. The initial printing involved finding the smallest layer height in combination with the fastest printing speed that was able to produce parts. These two parameters are necessary to be determined since a smaller layer height aids in the final surface finish and a fast printing speed keeps production time low. After stable prints could be made, a temperature sweep was started. The parameters determined from the initial printing were held fixed, while the temperature was changed in increments of 10 °C, within the range of 100 to 220 °C. The lower limit was determined from the melting temperature of the polymers and the upper temperature limit was defined by the 3D printer. Printing would continue until the entire range was covered or until the prints would fail at a given temperature. After each ring was printed, its geometry was measured using calipers. The outer diameter, wall thickness, and overall height were measured and recorded (estimated accuracy of the caliper, 50 μm). Each geometric feature was measured at five different locations along one ring. The measurements were averaged and used to evaluate the temperature effects. Other qualitative properties for each ring were also taken note of, such as surface quality, build plate adherence, and warping.

The density of the printed rings was measured and compared to the theoretical value calculated from the composition of the feedstock and the density of the single components. The density was determined with an analytical balance (LA 230 S, Sartorius, Göttingen, Germany) on the basis of Archimedes’ principle. First, after calibration using a ball bearing with a known mass, volume, and density, a ring sample was placed on the balance, and its weight was measured first in air and afterward in the fluid. Then, the balance device output the density of the given ring.

The roughness was measured with the MarSurf SD 26 (Mahr, GmbH, Göttingen, Germany) via a contact probe. The roughness values *R_a_* (average roughness around the mean line of the tested area) and *R_z_* (average of differences of maximum and minimum in test intervals) were determined at five separate locations on the ring and averaged. The probe was dragged 1.5 mm along the surface of the ring from the base of the print to the top surface at a velocity of 0.1 mm/s.

The rheological experiments using the binder and the feedstock materials were performed using a rotational rheometer (MCR 502, Anton Paar, Graz, Austria) in oscillatory mode. A plate–plate geometry with a sample diameter of 8 mm (PLA) or 14 mm (binder and feedstocks) was used, and a nitrogen atmosphere was applied. For temperature equilibration, a waiting time of 8 min was chosen. First, a strain sweep in the range of 0.001% to 10% at an angular frequency of 10 rad/s was performed in order to determine the linear viscoelastic range. Then, a frequency sweep in the range from 0.01 to 100 rad/s (starting at the highest angular frequency) at constant strain amplitude γ_0_ was carried out.

## 3. Results and Discussion

### 3.1. Physical Properties of Components and Feedstocks

First, the morphological and thermal properties of the feedstocks and its components are discussed. Basically, the shape, size, surface morphology, and volume fraction of the metallic particles influence the processing properties in fused filament fabrication and in sintering and, thus, the quality of the final metallic implant. Figure 3 shows a scanning electron micrograph of the titanium powder particles. The micrograph reveals that the Ti particles were mostly spherical as expected, since they were created by plasma atomization. The diameter of the Ti particles was smaller than 20 μm, and a quite broad size distribution can be observed using the production method. Agglomeration appeared to play only a minor role for these particles. Generally, a high powder loading in the feedstock (in the range of 50 to 65 vol.%) is necessary for sintering of metallic parts to facilitate homogeneous shrinkage, low distortion, and a high final density.

Since processing takes place at elevated temperatures, the thermal stability of the materials in this study is of high relevance and was investigated by thermogravimetric analysis. Furthermore, this method also gives information about the true weight fraction of the metallic particles of the prepared feedstocks. The results of the thermogravimetric investigations are shown in Figure 4 and Figure 5. Figure 4a depicts the relative mass as a function of temperature of the polymers and stearic acid, while Figure 4b depicts the derivative of these curves. The data reveal that thermal decomposition (significant increase of |*dm*/*dT*|) mainly took place above 160 °C for stearic acid and above 300 °C for the polymers. Consequently, the low-molecular-weight stearic acid was determined to be less thermally stable than the polymers. In conclusion, the thermal stability of the pristine components was mainly fulfilled for the purpose of the FFF process in this study, where only a short temperature interval with a maximum of 200 °C is sufficient.

In Figure 5a, the relative mass loss of the different feedstock materials is shown. The constant value of the relative mass at high temperatures corresponds to the weight fraction of the metallic particles. The data confirm the monotonic variation of feedstock concentration as intended. The data also reveal that the degradation temperature (peak of temperature derivative, as shown in Figure 5b) was not influenced by the particle concentration.

The quality of mixing could be evaluated using scanning electron micrographs of the feedstocks. Figure 6a,b display the morphology of the feedstocks with 50 and 60 vol.% titanium particles, respectively. Both feedstocks show a similar morphology with spherical titanium particles dispersed in the matrix of the binder material. The average particle diameter was about 20 μm (*D*_50_ = 13 μm and *D*_90_ = 20 μm for the probability distribution of the particle diameter *D* according to the supplier). This rough morphological analysis reveals that a spatially uniform distribution of particles was achieved. The neighbor-to-neighbor distance was quite small because of the high filling degree of these two feedstocks.

Differential scanning calorimetric investigations were carried out in order to determine the thermal transitions of the components of the binder, since processing could take place only above the highest melting temperature *T_m_*. The results of the DSC measurements are listed in Table 2. Stearic acid melts at a temperature range around *T_m_* = 58 °C. Lupolen V 2920 K has a peak melting temperature of 99 °C, and its melting range ends at around 105 °C. Vistamaxx™ 8880 is characterized by a similar melting temperature (*T_m_* = 97 °C), but has a larger melting interval which ends at around 115 °C. The glass transition of poly(isobutylene) is *T_g_* = −69 °C and consequently much lower.

The thermal transitions of the materials for FFF (PLA and the series of feedstocks) were also determined. Polylactide is characterized by a melting temperature of 150 °C and a broad crystallization interval with its maximum at 125 °C. The results of the DSC investigations of the in-house prepared feedstocks are shown in Figure 7. Figure 7a shows the second heating interval. The melting peaks of Lupolen V 2920 K and Vistamaxx 8880 superposed to a single peak. The cooling interval of the DSC investigations is presented in Figure 7b. Interestingly, the crystallization of stearic acid was moved to lower temperatures in the binder. In Figure 7c, the melting and crystallization temperatures as a function of titanium concentration are shown. The graphs reveal that the melting temperature decreased and the crystallization temperature increased with titanium loading until a saturation effect was achieved. This trend can be explained by the increase in thermal conductivity and heat capacity with an increasing concentration of metallic particles. This led to a different temperature hysteresis for DSC measurements in the material temperature at constant heating rate.

### 3.2. Rheological Properties

In this study, linear viscoelastic shear oscillations were performed, which allowed separating the elastic and viscous contributions of the material. This type of test is also suitable for elucidating the effect of low and high shear rates. In contrast to the capillary experiments performed in the work of Singh et al. [27], the flow properties at low shear rates were also investigated. These low shear rates become relevant during the layering process. The temperature was varied in order to investigate the change in yield stress with processing temperature.

A first hint of the rheological performance is given by the analysis of strain sweeps, which are generally performed in order to determine the linear regime. These experiments were performed at an angular frequency ω of 10 rad/s. The results are shown in Figure 8 for the polylactide used in this study and selected feedstocks. Polylactide is a linear homopolymer and is characterized by a linear range of at least up to 10% at the measurement temperature of 200 °C (Figure 8a). A different behavior is shown for the feedstocks. Figure 8b presents the dynamic moduli of selected feedstocks as a function of strain amplitude. The data of the dynamic moduli propose a clear trend. The linear range of the highly filled feedstocks was very small. An increasing filler concentration led to larger values of the moduli *G′* and *G″*. At higher strain amplitudes, the dynamic moduli of the highly filled feedstocks decreased. In addition, a crossover from a slightly more elastic behavior (*G′* > *G″*) to a slightly more viscous behavior (*G′* < *G″*) at the chosen frequency took place. This behavior can be explained by the presence of particle–particle interactions in the highly filled composites which break up at higher deformations. The strain sweep data reveal the relatively high dynamic moduli of highly filled composites and their strong dependence on applied deformation. It is also obvious that the linear viscoelastic range (strain interval where both moduli do not depend on strain amplitude) became smaller with higher particle loading. This effect was caused by the filler–filler interaction (friction and contact between the metal particles) which appears with a higher frequency for a higher filler loading. A larger particle concentration led to a higher number of filler–filler interactions which broke up at a given strain amplitude. In contrast to the polymer matrix of the polymer–ceramic composites studied in [29], in this work, the dynamic moduli of the pristine polymeric binder were much smaller at the chosen measurement temperature.

Frequency sweeps were performed at two different strain amplitudes and two different temperatures in order to demonstrate the specific viscoelastic nature of the feedstocks. These data allow one to analyze the response of the materials at different frequencies. The dynamic moduli of PLA are shown in Figure 9a. At the chosen test temperature, this grade of PLA had a low viscosity and attained only negligible values of storage modulus. Furthermore, it was also characterized by a large Newtonian plateau with a magnitude of complex viscosity (not shown). The role of filler concentration can be clearly seen in Figure 9b which shows storage and loss modulus as a function of titanium fraction for the in-house prepared feedstock materials. Both dynamic moduli increased with filler concentration. The frequency dependence of the moduli of the feedstocks with a high Ti loading depicted a pattern which significantly differed from the Maxwell model. The filler–filler interactions strongly contributed to the dynamic moduli and led to this specific behavior. In particular, at low frequencies, where the measured torque and stress were low, the particle–particle interactions strongly contributed to the moduli. At these low frequencies, the mechanical stress in the polymer matrix was very low, and the stress caused by the particle–particle interactions was much higher than the mechanical stress caused by the binder matrix. This mechanism led to a generally decreasing dependency of the dynamic moduli on the frequency of the powder-filled feedstocks.

Figure 10a,b present the magnitude of complex viscosity of the feedstocks as a function of angular frequency for two different temperatures. The slope of the curves became more negative with increasing powder concentration. Generally, a higher temperature yielded a lower viscosity. However, this trend was more pronounced at lower filler loading where the contribution of the polymer was significant in comparison to the feedstocks with a larger powder concentration where the filler–filler interaction dominated. It seems that the filler–filler interactions obeyed a different temperature dependence than the pristine polymers. This effect is also demonstrated by the data in Figure 11, which shows for two titanium concentrations the temperature dependence of the magnitude of complex viscosity. The temperature dependence of the feedstock with 60 vol.% powder particles was less pronounced than that with 50 vol.% powder loading.

Generally, a low or moderate viscosity of the feedstock is beneficial for FFF printing, whereas a too elastic behavior may be detrimental. Polylactide can be easily processed, but processing of highly filled feedstocks is more difficult. Increasing the powder concentration in the feedstocks raised the viscosity. A low value of viscosity in combination with a Newtonian or pseudo-plastic behavior caused the surface tension to create a smooth surface. In order to reduce the viscosity, the processing temperature may be increased if the melt is thermally stable and no significant degradation starts. However, our rheological data show that raising the processing temperature only yielded a limited viscosity reduction for the highly filled feedstocks because of the filler–filler interactions.

In this work, initial printing trials were used to determine how important printing parameters needed to change in order to achieve printed green bodies, such as printing speed and layer height. A temperature sweep was used to determine the admissible temperature range. A main parameter for processing is given by the concentration of metal powder. Figure 12 and Figure 13 present the influence of titanium loading on three principal processing parameters. The minimum processing temperature as experienced in our trials increased with powder loading (see Figure 12). Figure 13 reveals that an increasing powder loading yielded a lower maximum print velocity (caused by the viscosity increase) and a higher minimum layer height. The maximum print velocity decreased linearly with titanium loading, which indicates that the maximum filler fraction was limited by the processing properties.

Generally, a higher processing temperature seems to facilitate processing, as indicated by the data in Figure 14. With all other printing parameters constant and only the temperature changing, the mass of the printed part increased with processing temperature for the feedstock with 30 vol.% titanium (Figure 14a). The roughness of the printed part generally decreased with temperature (see Figure 14b). A simple equation was used that combined two aspects of the printed ring to evaluate the print deviation by means of a parameter *p_quality_*, which allows one to evaluate prints from quantifiable data.
(1)$$p_{quality}=\frac{1}{2}\left[\left|\frac{V_{real}-V_{ideal}}{V_{ideal}}\right|+\left|\frac{R_{a,printed}}{R_{a,filament}}\right|\right].$$

Equation (1) consists of adding the relative differences of the real and the set (ideal) volume (*V_real_* and *V_ideal_*) and of the roughness parameter *R_a_* of the printed part and the extruded filament (*R_a,printed_* and *R_a,filament_*), thus giving an estimation of the achieved geometric features of the printed part with an equal weight by volume and roughness. The ideal value for the volume was determined from the model volume while the optimum value for the roughness was assumed to be the roughness of the filament used to print the model. The addition of these relative values of printed volume and actual roughness resulted in Equation (1), quantifying the deviation from what was expected. A lower value denotes better representation of the model. Figure 14c shows the results of the analysis for selected feedstocks. Generally, the print quality parameter decreased with temperature, which indicates an improvement in the printed part for the feedstocks with a powder concentration up to 60 vol.%. Interestingly, the feedstock with 30 vol.% Ti particles was associated with the lowest values of *p_quality_*. This effect can be explained by an optimum viscosity which, on the one hand, was high enough in order to avoid large geometric distortions and, on the other hand, was low enough in order to achieve a smooth surface of the printed part. However, a powder concentration of 30 vol.% was insufficient for subsequent sintering, which shows that a compromise between printing and sintering properties has to be done.

The processing temperature and the particle concentration had a strong effect on the welding behavior. Figure 15 presents radiographs of a part of the printed ring for the feedstocks with 50 and 60 vol.% titanium particles at different temperatures. A qualitative evaluation of the tomographs is shown using a color gradient. At the lower temperatures, the layers were clearly visible because of contrast between adjacent layers; however, at high temperatures, the gradient became smaller and the wall showed a single shade of dark gray. Figure 15a,b reveal the influence of processing temperature. A higher processing temperature yielded a more complete fusion of deposited layers because of the lower viscosity of the polymer matrix. Increasing the particle concentration (Figure 15c) caused a higher viscosity which hindered the welding process. This impeded welding could be anticipated from the visually distinct deposited layers, i.e., by the larger contrast of light and dark. Whereas a high concentration of powder particles was favored for sintering, a higher filler loading led to a more narrow range of processing parameters. In addition, the maximum filler loading of the feedstock was associated with blocking of particles in the nozzle. This result also underlines that one has to find a balance between processing properties and sinterability.

## 4. Conclusions

In this work, the rheological properties of several polymer–metal feedstocks for sintering of metallic parts were investigated and related to their processing properties by means of fused filament fabrication (FFF). In the FFF process, a low or moderately viscous binder system was combined with a high volume fraction of metal particles. The high filler concentration, which was close to the theoretical maximum filler fraction, gave rise to a peculiar rheological behavior, i.e., a solid-like behavior at low shear rates, which influenced the FFF behavior. In contrast to, e.g., the homopolymer polylactide which predominantly behaved in a Newtonian manner at the chosen printing temperature, the commercial and in-house designed feedstock materials were characterized by a pronounced (apparent) yield stress which led to a more narrow range of processing parameters. A minimum printing temperature existed which had to be exceeded in order to achieve satisfying printing results. For the chosen low-viscosity binder, the viscosity and the yield stress only moderately changed with temperature for the feedstocks with a high powder loading. The viscosity of the polymeric component in the binder decreased with temperature. However, at a high particle loading, the particle–particle interactions equally contributed to the viscosity and dampened the temperature effect of the feedstocks. A higher printing temperature led to a smoother surface of the printed part and a more complete welding of deposited layers (smaller roughness). The existence of an apparent yield stress limited the maximum particle concentration. It hindered the formation of smooth surfaces, slowed down welding of deposited layers, and limited the maximum printing velocity because of the high viscosity of highly filled polymer–metal composites. Since a minimum powder fraction was necessary for the subsequent sintering process, an optimum powder concentration existed where the viscosity in the relevant range of shear rates was low enough in order to warrant a sufficient printing quality by maintaining the possibility to achieve high-quality sintered metallic parts.

## Figures and Tables

**Figure 1 materials-13-04413-f001:**
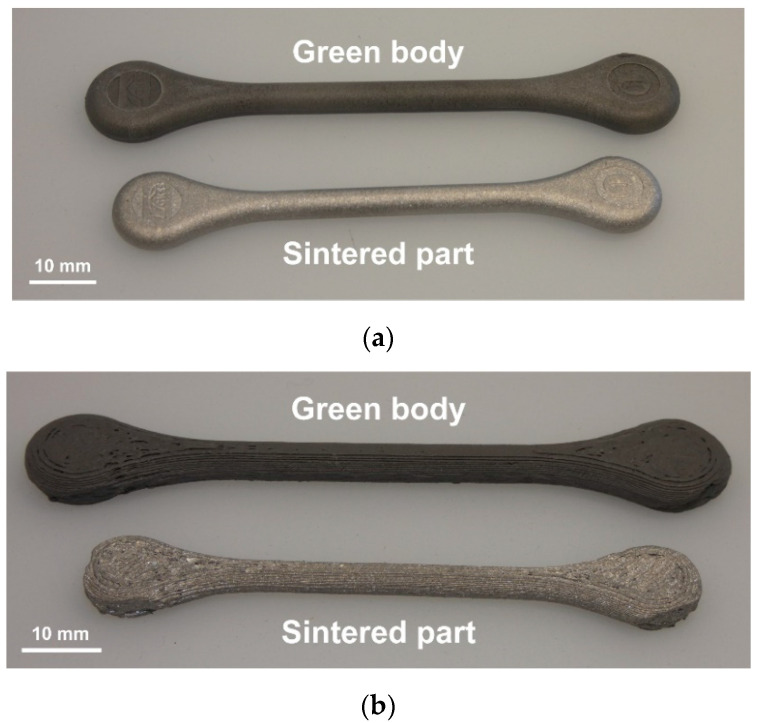
Green body and sintered titanium tensile bar using (**a**) metal injection molding (MIM) and (**b**) fused filament fabrication (FFF). The length of the injection-molded (MIM) tensile bar is 89 mm.

**Figure 2 materials-13-04413-f002:**
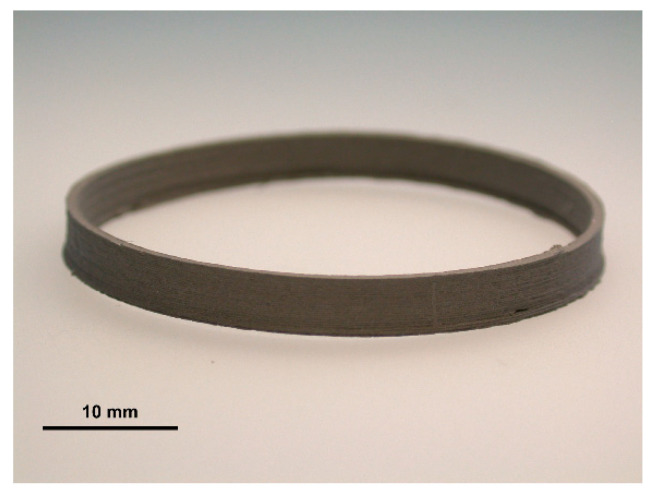
Photograph of printed ring using the in-house designed feedstock. The diameter of the ring is 40 mm.

**Figure 3 materials-13-04413-f003:**
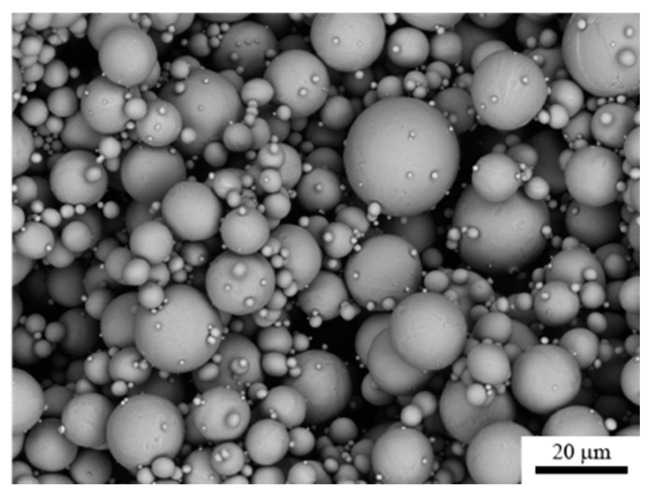
Scanning electron micrograph of Ti particles of this study.

**Figure 4 materials-13-04413-f004:**
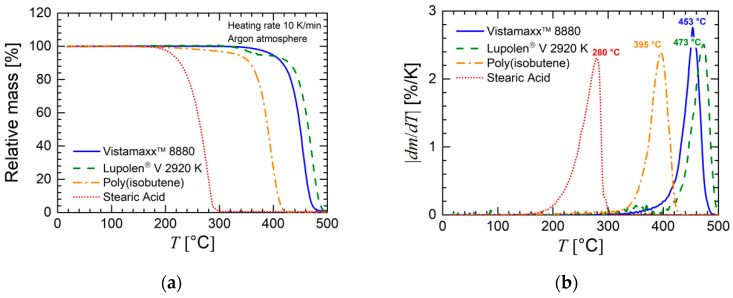
(**a**) Relative mass and (**b**) derivative of mass loss (*dm*/*dT*) as a function of temperature *T* determined by thermal gravimetric analysis for the polymers (feedstock components, see Table 1) of this study and stearic acid. The measurements were performed in an argon atmosphere. The heating rate was 10 K/min.

**Figure 5 materials-13-04413-f005:**
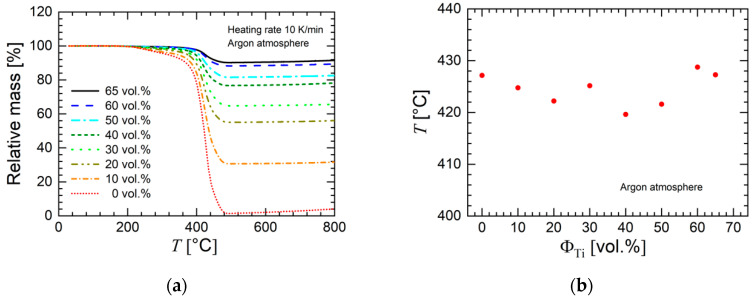
(**a**) Relative mass as a function of temperature determined by thermal gravimetric analysis for the prepared feedstocks of this study (argon atmosphere). The heating rate was 10 K/min. (**b**) Temperature at maximum degradation (peak temperature obtained by the derivative of relative mass loss) as a function of metal powder concentration Φ_Ti_.

**Figure 6 materials-13-04413-f006:**
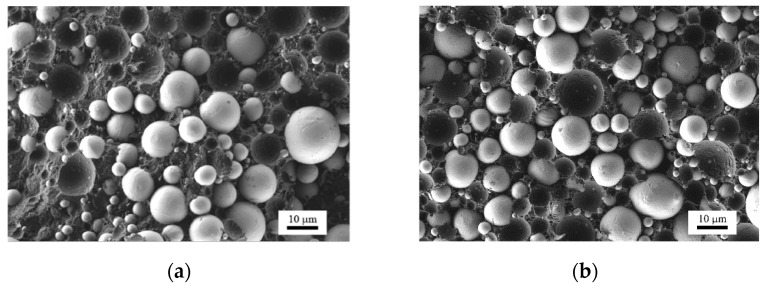
Scanning electron micrographs of compression-molded samples of (**a**) the Ti feedstock with 50 vol.% metal loading and (**b**) the Ti feedstock with 60 vol.% metal loading.

**Figure 7 materials-13-04413-f007:**
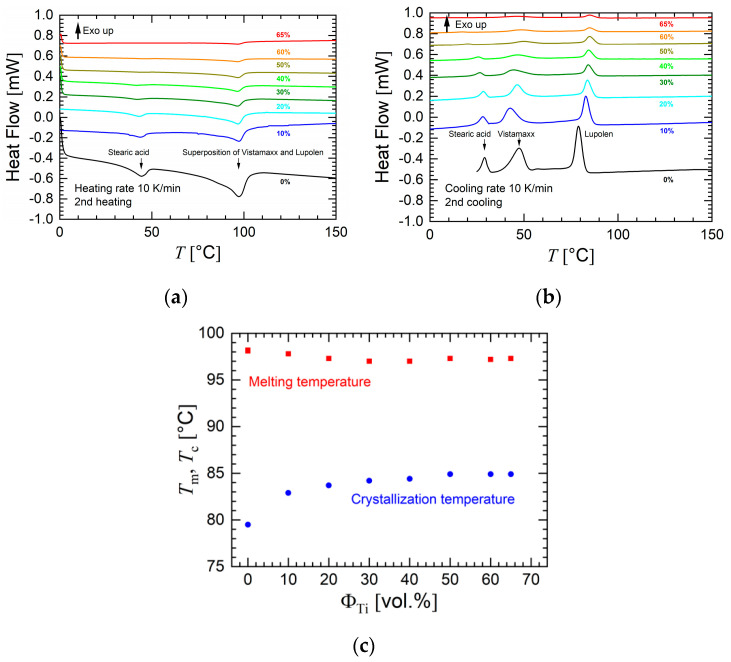
Results of differential scanning calorimetry investigations in nitrogen atmosphere. The heating and cooling rates were 10 K/min. (**a**) Second heating interval. (**b**) First cooling interval. (**c**) Highest melting and crystallization temperatures (*T_m_*, *T_c_*) as a function of metal concentration. The data in (**a**,**b**) were shifted for clarity.

**Figure 8 materials-13-04413-f008:**
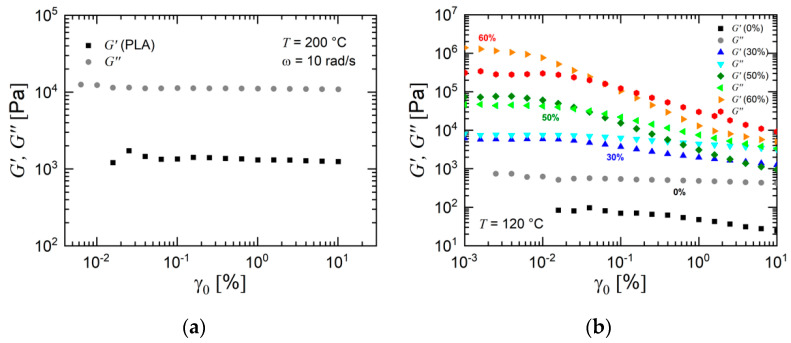
Strain sweeps at an angular frequency of ω = 10 rad/s for a polylactide and selected feedstocks of this study. (**a**) Polylactide at a temperature of 200 °C and (**b**) selected feedstocks at 120 °C.

**Figure 9 materials-13-04413-f009:**
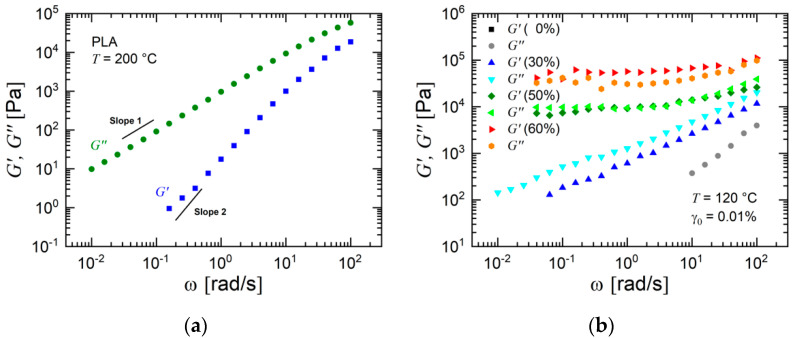
Storage modulus *G′* and loss modulus *G″* of (**a**) polylactide and (**b**) selected feedstocks as a function of angular frequency ω. The shear amplitude was in the linear range with (**a**) γ_0_ = 5% and (**b**) γ_0_ = 0.01%. In (**a**), the slopes indicate the terminal behavior of the Maxwell model.

**Figure 10 materials-13-04413-f010:**
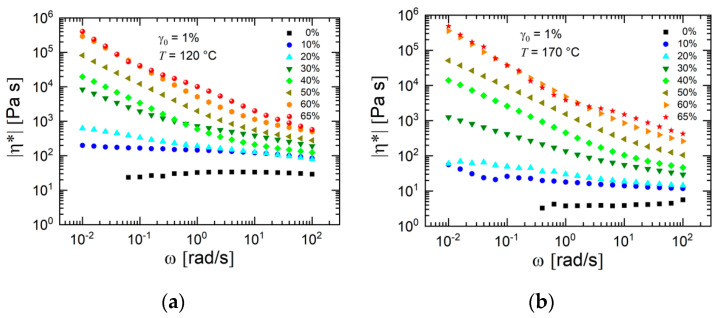
Magnitude of complex viscosity η* of the prepared feedstocks at a test temperature of (**a**) 120 °C and (**b**) 170 °C. The shear amplitude γ_0_ was equal to 1%.

**Figure 11 materials-13-04413-f011:**
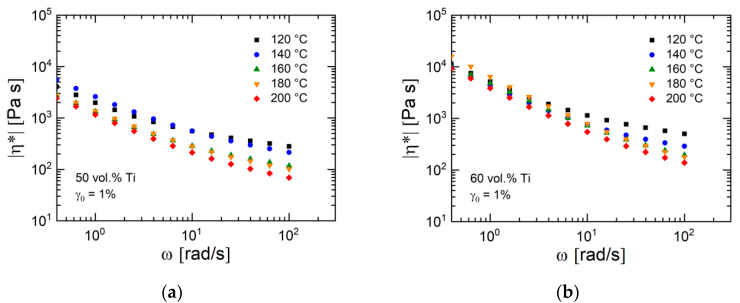
Magnitude of complex viscosity η* for the feedstocks with (**a**) 50 and (**b**) 60 vol.% Ti particles as a function of angular frequency ω at five temperatures. The shear amplitude γ_0_ was 1%.

**Figure 12 materials-13-04413-f012:**
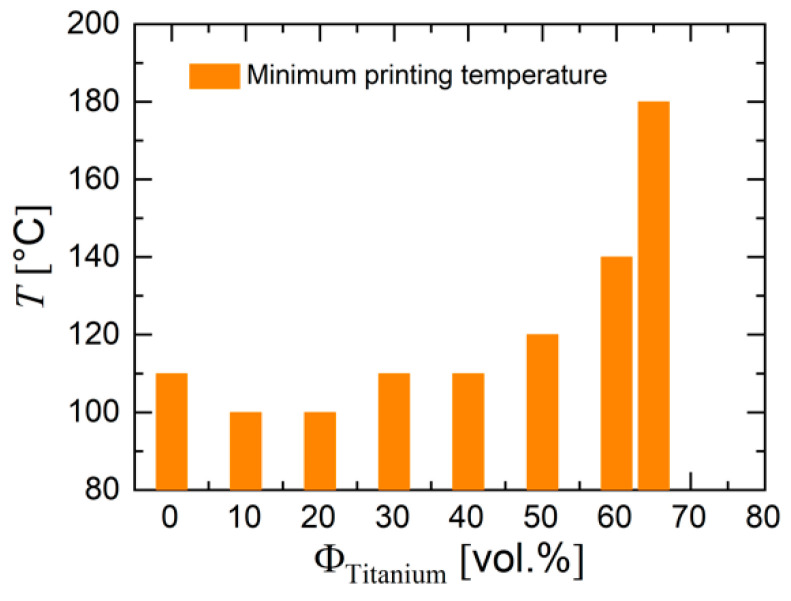
Influence of powder concentration on minimum printing temperature.

**Figure 13 materials-13-04413-f013:**
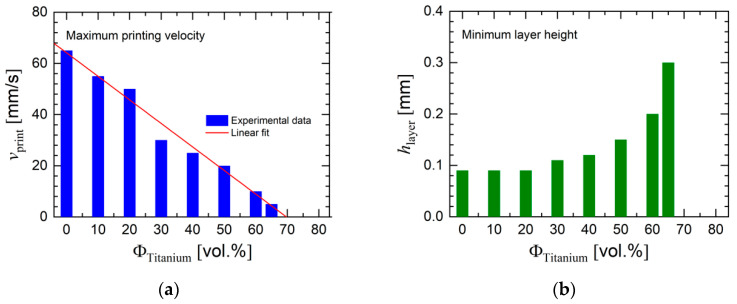
Influence of powder concentration on (**a**) printing velocity and (**b**) minimum layer height for the series of titanium feedstocks.

**Figure 14 materials-13-04413-f014:**
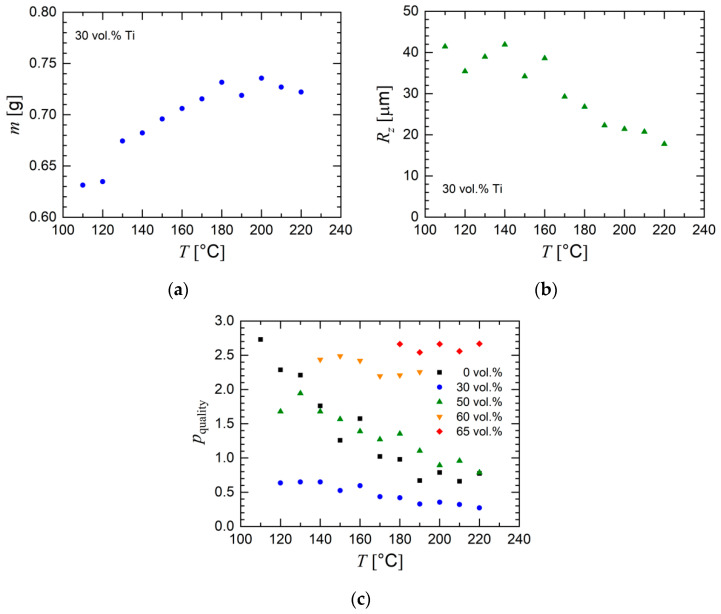
Influence of printing temperature on the quality of the printed green body. (**a**) Mass *m* and (**b**) roughness *R_z_* of the printed part for the feedstock with 30 vol.% titanium particles. (**c**) Value of the quality parameter (Equation (1)) as a function of temperature. The powder concentration is indicated.

**Figure 15 materials-13-04413-f015:**
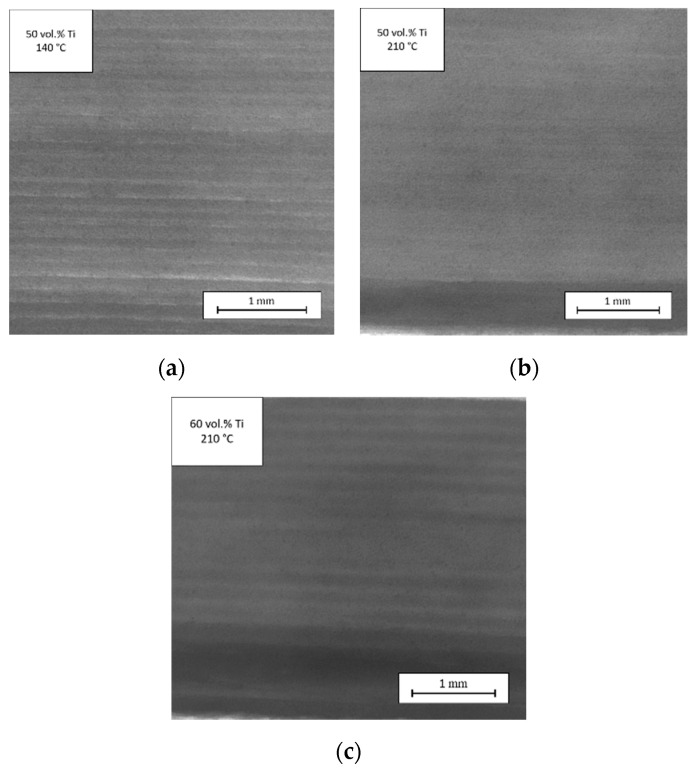
Influence of temperature and titanium concentration on welding of deposited layers: (**a**) 50 vol.% Ti particles and processing temperature of 140 °C; (**b**) 50 vol.% Ti particles and processing temperature of 210 °C; (**c**) 60 vol.% Ti particles and processing temperature of 210 °C.

**Table 1 materials-13-04413-t001:** Composition of the binder for the in-house prepared feedstocks.

Component	Function	Composition (vol.%)	Composition (wt.%)
Poly(ethylene–vinyl acetate) copolymer (Lupolen^®^ V 2920 K)	Backbone	26.8	27.5
Poly(propylene–ethylene) copolymer (Vistamaxx™ 8880)	Main	46.3	45.0
Poly(isobutene)	Main	22.1	22.5
Stearic acid	Additive	4.8	5.0

**Table 2 materials-13-04413-t002:** Physical properties of polylactide (PLA) and the binder components.

Polymer	Supplier	Density ^1^ at Room Temperature(g/cm^3^)	*T_g_* (°C)	*T_m_* (°C)	*T_c_* (°C)
Polylactide (PLA)	REC	1.24	60	150	125
Poly(ethylene–vinyl acetate) Copolymer (Lupolen^®^ V 2920 K)	LyondellBasell	0.93	^2^	99	79
Poly(propylene–ethylene) Copolymer (Vistamaxx™ 8880)	ExxonMobil	0.88	−21	97	47
Poly(isobutene)	Sigma-Aldrich	0.92	−69	-	-
Stearic acid	Merck	0.94	-	58	52

^1^ The density values are taken from the data sheets of the suppliers. ^2^ The glass transition could not be determined.

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
