# Peer review of "Processing of Highly Filled Polymer–Metal Feedstocks for Fused Filament Fabrication and the Production of Metallic Implants"

_materials, 2020, doi:10.3390/ma13194413_

Round 1
Reviewer 1 Report
My comments are general and apply to the entire text of the manuscript.
The presented studies are important in the context of their application. The authors present their research in relation to implants, but do not provide their specific application and the considered geometry of these implants. The given example of the ring structure analysis in no way reflects the applicability of the results of these tests. The described study could be implemented in the form of flowcharts, which would improve readability. The main aim of the research was stated, but the authors do not state what the purpose of each stage of the research was.
Sample questions:
How was shrinkage eliminated?
2. redundant references to footnotes without explaining the essence of the problem in the quoted manuscript
3. Is the caliper the correct measuring tool - what is the accuracy of the measurement? Ring geometry measurement ??? diameter? Wall thickness?
4. Incorrect statement: scale measurement - mass measurement correct !!
In many places the description is illegible and requires editing.
The quality of all the drawings is questionable and requires significant improvement.
Author Response
Reviewer #1:
- “The authors present their research in relation to implants, but do not provide their specific application and the considered geometry of these implants. The given example of the ring structure analysis in no way reflects the applicability of the results of these tests. The described study could be implemented in the form of flowcharts, which would improve readability. The main aim of the research was stated, but the authors do not state what the purpose of each stage of the research was.”
- We now mention in more detail the relevance to medical implants (lines 128 - 130 on p. 3, lines 203 - 206 on p. 5) and have motivated the experiments in Sections 2 and 3 (see the new passages in yellow color). We dispensed with the insertion of a flow chart (although it is a good idea) because of the high number of figures. The additional explaining text helps to understand the study (see also comment 5 of reviewer 2).
- “How was shrinkage eliminated ?”
- Shrinkage mainly refers to the sintering process which was not carried out in this study. In practice, the dimension of the green part is scaled by the shrinkage factor, which is determined experimentally.
- “Redundant references to footnotes without explaining the essence of the problem in the quoted manuscript.”
- The content of some more references have been explained, see pp. 3.
- “Is the caliper the correct measuring tool - what is the accuracy of the measurement?”
- The systematical error of the caliper is now given on line 221 (p. 6) of the revised manuscript.
- “Incorrect statement: scale measurement - mass measurement correct”
- The word “scale” has been replaced in lines 227 to 230 on p. 6.
- “The quality of all the drawings is questionable and requires significant improvement.”
- Size and quality of the figures are increased now.
Reviewer 2 Report
In this work the authors analysis the process of highly filled polymer-metal feedstocks for fused filament fabrication and the production of Metallic Implants. The research appears to be efficiently done and appropriately reported, however the standard of English is acceptable only needs few corrections. Nevertheless, there some questions and corrections that must be answered to improve and complete the document.
Line 22. Please change “Small amplitude shear oscillations reveal …” to “Small amplitude of shear oscillations reveal …”
Introduction section: In this section the authors don’t indicate the novelty of their work. what is the innovation of your work when compared with the other researchers? The "Knowledge gap to be filled"? In this introduction the authors must describe or indicate the work that will be done to test their "hypothesis".
Line 46. The first time the abbreviations you must indicate meaning of it. Please refer the meaning of “MIM” in this line.
Lines 86, 88, 96. It is preferable to indicate the name of the authors and after te reference number. Example for line 86: … given by Turner et al. [18] …
The section 2. could be more complete. In this section, authors should write a paragraph where they briefly describe what they are going to do experimentally. As it is now, the authors present what they are going to do, point by point, but without a clear link between the different points. Another improvement that I suggest for this section is to complete some descriptions with images of experimental work done in the laboratory. For example, it would be interesting if the authors show an image of the rheological samples, among other images. It is very important for the reader to understand clearly what was done. You must remember that you are present a scientific work that only could be validate if other researchers could repeat your work and obtain the same results.
Lines 353-354, the authors claim that “thermal decomposition takes place above 160ºC for stearic acid and above 300 °C for the polymers”. However, consulting the Figure 4 it is not very clear these values. Please, can you clarify the methodology that you used to achieve these values?
Lines 300-306. All this paragraph is in bold, please correct the formatting.
The Figures 8, 9 and 15. Must be improved because they are not very clear, and resolution is very low.
Author Response
Reviewer #2:
- “Line 23. Change “Small amplitude shear oscillations …” to “Small amplitude of shear oscillations …”
- The statement was changed in the abstract (p. 1).
- Introduction: In this section the authors don’t indicate the novelty of their work. What is the innovation of your work when compared with the other researchers? The "Knowledge gap to be filled"? In this introduction the authors must describe or indicate the work that will be done to test their "hypothesis".”
- We have inserted on p. 3 the statement that this work provides a first step towards more understanding by aiming to link rheological and printing properties of feedstocks which are of relevance for medical implants. Therefore thermal analysis, rheological experiments and printing were performed in this work.
- “Line 48. The first time the abbreviations you must indicate meaning of it.”
- The abbreviation “MIM” has been introduced in this line.
- “Lines 86, 88, 96. It is preferable to indicate the name of the authors and after the reference number. Example for line 86: … given by Turner et al. [18]”
- The author’s names have been inserted on several lines on p. 3 of the revised manuscript.
- “In this section (2.), authors should write a paragraph where they briefly describe what they are going to do experimentally. As it is now, the authors present what they are going to do, point by point, but without a clear link between the different points. Another improvement that I suggest for this section is to complete some descriptions with images of experimental work done in the laboratory. For example, it would be interesting if the authors show an image of the rheological samples, among other images.”
- The experiments are now motivated in Sections 2 and 3, see pp. 4 to 11.
- “Lines 353 - 354: The authors claim that “thermal decomposition takes place above 160 °C for stearic acid and above 300 °C for the polymers”. However, consulting the Figure 4 it is not very clear these values.”
- These two temperatures (160 and 300 °C) correspond to the lower border of the degradation peak shown in Figure 4(b). We have inserted the term “significant increase of |dm/dT| in lines 274 and 275 on p. 8.
- “Lines 300 - 306. All this paragraph is in bold, please correct the formatting.”
- The bold font has been eliminated in the revised manuscript.
- “The Figures 8, 9 and 15. Must be improved because they are not very clear, and resolution is very low.”
The quality of the figures has been improved.
Reviewer 3 Report
This paper investigates the rheological behaviors of various polymer-metal feedstocks for the fused filament fabrication process. I think this is an interesting study to reveal the flow properties of feedstock materials for optimum processing, which would have a significant impact on the sintered parts. The authors conducted sufficient experimental results and detailed analysis, and the paper is written in a logical order. A few questions are listed below.
- I would recommend putting the scale bars in Figures 1 and 2.
- Please specify the detectors (BSE or SE) used for Figure 3. It looks like there are some satellites on Ti particles, are they taken account into the particle diameter calculation?
- Same for Figure 6, please specify if it is a BSE or SE image.
- I would recommend the authors to adjust the Figure/legend size. For example, the legend of Figure 8(b) is hard to read even though it is zoomed in at 150%.
- Figure 15, could you explain how to determine the complete fusion of deposited layers? Quantitively?
Author Response
Reviewer #3:
- “I would recommend putting the scale bars in Figures 1 and 2.”
- Figures 1 and 2 now include the scale bars.
- “Please specify the detectors (BSE or SE) used for Figure 3. It looks like there are some satellites on Ti particles, are they taken account into the particle diameter calculation?”
- The particle diameter distribution was determined by the supplier and is supposed to include all particles (p. 8, line 288). The detectors are now mentioned in the revised paper (pp. 5 and 6, lines 173 and 178).
- “I would recommend the authors to adjust the Figure/legend size. For example, the legend of Figure 8(b) is hard to read even though it is zoomed in at 150%.”
- The size and quality of the figures were increased. (Change from tif files to png files).
- “Fig. 15: Could you explain how to determine the complete fusion of deposited layers? Quantitatively?”
Our qualitative evaluation of the tomographs is based on the color gradient. At lower temperatures the layers are clearly visible with a sharp contrast between adjacent layers. At higher temperatures, the gradient becomes smaller, see the new lines 441 to 444. A quantitative evaluation would be too elaborated.
Round 2
Reviewer 1 Report
Thank you for considering my comments and suggestions.
I hope that it will be the beginning of a series of publications on this topic, and the authors will base their further research on specific medical devices.
Reviewer 2 Report
The second version of manuscript improved significantly when compared with first version. So, in my opinion the manuscript can be accepted for publication.